# Determination of n-alkanes, PAHs and hopanes in atmospheric aerosol: evaluation and comparison of thermal desorption GC-MS and solvent extraction GC-MS approaches

Meng Wang[1, 2, 3], Ru-Jin Huang[1, 2, 4], Junji Cao[1, 2, 4], Wenting Dai[1, 2], Jiamao Zhou[1, 2], Chunshui Lin[1, 2], Haiyan, Ni[1, 2], Jing Duan[1, 2], Ting Wang[1, 2], Yang Chen[5], Yongjie Li[6], Qi Chen[7], Imad El Haddad[8] and Thorsten Hoffmann[9]

[1]State Key Laboratory of Loess and Quaternary Geology (SKLLQG), Institute of Earth Environment, Chinese Academy of Sciences, Xi'an 710061, China
[2]Key Laboratory of Aerosol Chemistry & Physics (KLACP), Institute of Earth Environment, Chinese Academy of Sciences, Xi'an 710061, China
[3] University of Chinese Academy of Sciences, Beijing 100049, China
[4] CAS Center for Excellence in Quaternary Science and Global Change, Xi'an, Chinese Academy of Sciences, 710061, China
[5]Key Laboratory of Reservoir Aquatic Environment of CAS, Chongqing Institute of Green and Intelligent Technology, Chinese Academy of Sciences, Chongqing 400714, China
[6]Department of Civil and Environmental Engineering, Faculty of Science and Technology, University of Macau, Taipa, Macau, China
[7]State Key Joint Laboratory of Environmental Simulation and Pollution Control, College of Environmental Sciences and Engineering, Peking University, Beijing 100871, China
[8]Laboratory of Atmospheric Chemistry, Paul Scherrer Institute, 5232 Villigen PSI, Switzerland
[9]Institute of Inorganic and Analytical Chemistry, Johannes Gutenberg University of Mainz, Duesbergweg 10–14, 55128 Mainz, Germany

*Correspondence to*: Ru-Jin Huang (rujin.huang@ieecas.cn)

**Abstract.** Organic aerosol (OA) constitutes a large fraction of fine particulate matter (PM) in the urban air. However, the chemical nature and sources of OA are not well constrained. Quantitative analysis of OA is essential for understanding the sources and atmospheric evolution of fine PM, which requires accurate quantification of some organic compounds (e.g., markers). In this study, two analytical approaches, i.e., thermal desorption (TD) gas chromatography-mass spectrometry (GC-MS) and solvent extract (SE) GC-MS were evaluated for the determination of n-alkanes, polycyclic aromatic hydrocarbons (PAHs), and hopanes in ambient aerosol. For the SE approach, the recovery obtained is 89.3–101.5%, the limits of detection (LOD) are 0.05–1.1 ng (1.5–33.9 ng m$^{-3}$), repeatability is 3.5–14.5% and reproducibility is 1.2–10.9%. For the TD approach, the recovery is 57.2–109.8%, the LODs are 0.1–1.9 ng (0.04–0.9 ng m$^{-3}$), repeatability is 2.1–19.4% and reproducibility is 1.1–12.9%. Ambient aerosol samples were collected from Beijing, Chengdu, Shanghai, and Guangzhou, during the winter of 2013 and were analyzed by the two methods. After considering the recoveries, the two methods show a good agreement with a high correlation coefficient ($R^2 > 0.98$) and a slope close to unity. The concentrations of n-alkanes, PAHs, and hopanes are found to be much higher in Beijing than those in Chengdu, Shanghai, and Guangzhou, most likely due to emissions from traffic and/or coal combustion for wintertime heating in Beijing.

## 1 Introduction

Fine particulate matter (with aerodynamic diameters $\leq 2.5$ μm, PM$_{2.5}$) has significant impacts on human health (Cao et al., 2012; Lelieveld et al., 2015; Zheng et al., 2015; Cohen et al., 2017), visibility (Lin et al., 2014; Shen et al., 2015) and climate (Kanakidou et al., 2005; Gustafson et al., 2011; IPCC, 2014). It is composed of a large variety of inorganic and organic compounds, in which the organic composition is much more complicated because of a very large chemical space with respect to molecular weight, functional groups and polarity (Zhang et al., 2007; Jimenez et al., 2009; Huang et al., 2014; Wang et al., 2018). Molecular-level characterization of the organic compounds in PM$_{2.5}$ is therefore essential for better understanding of the chemical nature, sources, atmospheric processes and impacts of PM$_{2.5}$ (Surratt et al., 2008; Chan et al., 2011; Fu et al.,

2016). Various efforts have been committed to study the composition of organic aerosol in different environments including the urban areas (Chan et al., 2013) and forested areas such as Amazon (Hu et al., 2015) and the southeastern United States (Zhang et al., 2018).However, in many environments, only 10–30% of the particulate organic matter has been identified as specific compounds despite years of effort and the use of the most sophisticated techniques available (Hoffmann et al., 2011).

The conventional approach for molecular analysis of organic compounds often requires the extraction of aerosol collected on filter into solvents followed by evaporation, separation, and detection by e.g. gas chromatography-mass spectrometry (GC-MS) (Kourtchev et al., 2009; Nozière et al., 2015; Maenhaut et al., 2017; Yu et al., 2017). However, solvent extraction (SE) method is prone to contamination and/or analyte lost during sample pre-treatment and is environmentally unfriendly because of the use of a large amount of solvent (Samy et al., 2010; Giri et al., 2013; Urban et al., 2014; Yang et al., 2017). In addition,

the SE method is often time-consuming. Thermal desorption (TD) is an alternative approach which uses an elevated temperature to evaporate organic analytes from the filter for subsequent GC-MS detection (Ho and Yu, 2004; Chow et al., 2007; Xu et al., 2013). Compared to SE, the TD method is convenient and solvent free (Graham et al., 2010; Lambe et al., 2010; Ramírez et al., 2010; Kim et al., 2016). The application of TD/GC-MS on the qualitative and quantitative analysis of ambient organic compounds such as polycyclic aromatic hydrocarbons (PAHs) in $PM_{2.5}$ samples has demonstrated its

analytical potential (Orecchio, 2011; Cheruiyot et al., 2015; Wang et al., 2015; Yang et al., 2017). Also, Ho and Yu (2004) compared the TD and SE methods for n-alkanes and PAHs with 16 ambient filter samples from Hong Kong; and Ho et al. (2008) compared these two methods for alkanes, PAHs, cyclohexanes, steranes, phthalates, and hopanes with 14 ambient samples from Hong Kong and for PAHs with 19 ambient samples from Tongliang. However, the evaluation and comparison of TD/GC-MS with the well-established SE/GC-MS are still needed to test its capability of measuring ambient samples of

low-to-high concentrations and different emission sources. Particulate air pollution has become a serious environmental problem in China over the past decades (Chan and Yao, 2008; Zhao et al., 2009; Zhang et al., 2012; Jiang et al., 2015). Understanding the chemical composition and sources of fine particles is therefore essential for mitigating particulate air pollution. This is particularly true for organic aerosol (OA) which often constitutes more than 50% of fine PM mass in urban areas during haze pollution events (Takegawa et al., 2006; Srivastava et al., 2018) and is much less constrained compared to

the inorganic fraction. In this study, we evaluated and compared the TD/GC-MS and SE/GC-MS methods in terms of their analytical performance including sensitivity, recovery, limit of detection (LOD), repeatability, and reproducibility. Then the two methods were applied for the determination of alkanes, PAHs, and hopanes in ambient aerosol samples collected in four Chinese megacities including Beijing, Shanghai, Chengdu, and Guangzhou, representing North, East, West, and South of China where the sources of these organics are different. The mass loadings of PM in those four cities were also quite different.

The main objective was to show that the TD/GC-MS method can be used to measure samples of different concentrations and different emission sources.

## 2 Experimental

### 2.1 Sampling

Twenty-four-hour integrated $PM_{2.5}$ samples were collected on pre-backed (780 °C, 3 h) quartz fiber filter (8 inch×10 inch)

using a high-volume sampler at a flow rate of 1.05 $m^3$ $min^{-1}$ from December 2013 to January 2014 in the four megacities in China (Table S1). Eleven to twelve filters were collected from each sampling site, including 2 field blanks. The filters were wrapped in aluminum foils and stored at –20 °C until laboratory analysis.

### 2.2 SE/GC-MS analysis

A portion (4.34 $cm^2$) of the sample or blank filter was cut with a clean stainless-steel punch, then extracted with a mixture (15

mL) of dichloromethane (DCM) (99.9%, LC grade, Mallinckrodt Laboratory Chemicals, Phillipsburg, NJ, USA) and methanol

(99.9%, LC grade, Mallinckrodt Laboratory Chemicals, Phillipsburg, NJ, USA) (3:1, v/v) under ultra-sonication for 15 min and filtered through quartz wool packed in a Pasteur pipette (Fujii et al., 2016; Mohseni Bandpi et al., 2017). The extraction procedure was repeated for 3 times. Then, the extracts were concentrated using a rotary evaporator under a vacuum condition and blown to < 0.5 mL with a gentle stream of pure nitrogen. The remaining solution was transferred to a 1.5 mL brown vial. Finally, 25 μL n-$C_{24}D_{50}$ (20 μg mL$^{-1}$) and 25 μL fluoranthene-$d_{10}$ (20 μg mL$^{-1}$) were added as co-injection internal standards to correct for the recovery during GC-MS measurements. Note that the suspended particles (if any) were removed during the filtration between extraction steps which, however, has little influence on the extraction efficiencies of the measured organics because the extraction solvents cover a large range of polarities and can efficiently dissolve n-alkane, PAHs and hopanes measured here as demonstrated in previous studies (Fujii et al., 2016). The high extraction efficiencies are further supported by the good recoveries (89.4−98.6%) which are considered to be very satisfactory given additional sources of error (e.g., from GC-MS detection). The final volume of the solution was fixed at 1.0 mL. For the analysis of standards, a mixture of standards was spiked onto the pre-cleaned blank filters and then followed the same procedure as for ambient samples. The extracts were kept in a refrigerator until GC-MS analysis and 1 μL was injected into the GC-MS system.

The GC-MS analysis was performed using an Agilent 7890 GC coupled with an Agilent 5975C mass spectrometer detector (MSD). The GC oven program was set at an initial value of 50 °C, held at this temperature for 1 min, programmed at a rate of 25 °C min$^{-1}$ to 140 °C and 10 °C min$^{-1}$ to 300 °C, and then held at the final temperature of 300 °C for 5 min. The column was an HP-5MS (5% diphenyl/95% dimethylsiloxane, 30 m×0.25 mm×0.25 μm). The carrier gas was helium which was held at a constant pressure of 7.7 psi. The MSD was operated at 280 °C and 70 eV for electron ionization. The scan range of mass-to-charge ratio (m/z) was from 50 to 650 atomic mass unit (amu).

**2.3 TD/GC-MS analysis**

The filter punch of 0.53 cm$^2$ was cut with the stainless-steel punch. Two internal standards 25 μL n-$C_{24}D_{50}$ (20 μg mL$^{-1}$) and 25 μL fluoranthene-$d_{10}$ (20 μg mL$^{-1}$) with dichloromethane were spiked onto the filter punch. After air-drying for a few seconds, the punch was divided into four roughly equal portions with a razor blade, then loaded into the TD tube. The TD tube is a Pyrex glass tube and is cleaned following a standard procedure: it was cleaned with methanol under ultra-sonication for more than 30 min, then baked at 550 °C for at least 10 hours before use (Ho and Yu, 2004; Ho et al., 2008; Xu et al., 2013; Wang et al., 2015). A small amount of pre-baked glass wool (baked at 550 °C for at least 5 h) was used to make two plugs for holding the filter pieces in the middle of the tube. After the loaded TD tube was placed in the injector port, the septum cap was closed and the injector-port temperature was raised from 50 °C to 280 °C for desorption (about 8–9 min). During this period, the GC oven temperature was initially at 30 °C, held at this temperature for 2 min, programmed at a rate of 20 °C min$^{-1}$ to 120 °C and 10 °C min$^{-1}$ to 300 °C, and then held at the final temperature of 300 °C for 10 min. The analysis time of each sample was about 52 minutes. The injector-port temperature was optimized by testing a set of temperature programs (see below). The injector was set in the splitless mode for the first 2 min in the GC temperature program, then switched to the split mode, and returned to the splitless mode at the end of the GC run. When the analysis was finished, the cooling program automatically started which took about 35 minutes. When the injection temperature was down to 50 °C, the next sample was put into the injector. A new TD tube was used for each analysis to avoid any potential contamination carry-over to the next analysis. The standard calibration curves were established for individual compounds by spiking standards of targeted compounds onto pre-cleaned filters which were then measured following the same procedure as for ambient samples. The GC column was an HP-5MS (5% diphenyl/95% dimethylsiloxane, 30 m×0.25 mm×0.25 μm). The carrier gas was ultra-high purity helium (99.9999%) which was held at a constant pressure of 7.7 psi and flow of 1.0 ml min$^{-1}$ (Ho and Yu, 2004; Ho et al., 2008). The MSD was operated at 280 °C and 70 eV for electron ionization. The mass scan range (m/z) was the same with the SE method (50–650 amu).

**2.4 Quality assurance and control (QA/QC)**

Twenty-four n-alkanes ($C_{14}$–$C_{37}$) (99%, Aldrich, Milwaukee, WI, USA), twelve PAHs (99%, Aldrich, Milwaukee, WI, USA (Acenaphthene (Ace), Fluorene (Fl), Phenanthrene (Phe), Anthracene (Ant), Fluoranthene (Fla), Pyrene (Pyr), Benz(a)anthracene (BaA), Chrysene (Chry), Benzo(b)fluoranthene (BbF), Benzo(k)fluoranthene (BkF), Benzo(a)pyrene (BaP), Indeno(1,2,3-cd)pyrene (IcdP), Dibenz(a,h)anthracene (DahA), Benzo(g,h,i)perylene (BghiP)) and eight hopanes (99%, Supelco, Bellefonte, PA, USA) (22,29,30-Trisnorphopane (TH), C29αβ-hopane (C29αβH), C29αα-+βα-hopane (C29αα-+βαH), C30αβ-hopane (C30αβH), C30αα-hopane (C30ααH), C30βα-hopane (C30βαH), C31αβS-hopane (C31αβSH), C31αβR-hopane (C31αβRH)) were quantified using both the TD and SE methods. Internal standards (n-$C_{24}D_{50}$ (98%, Aldrich, Milwaukee, WI, USA) and fluoranthene-$d_{10}$ (98%, Aldrich, Milwaukee, WI, USA)) were added during the pre-treatment process for two methods. Direct injection of a mixture of liquid standards with different concentrations into the SE/GC-MS system was utilized to establish a calibration curve, while a mixture of standards with different concentrations were spiked onto the pre-cleaned filters and then used to establish the calibration curves for the TD/GC-MS system. Table S2 shows slope and the correlation coefficients ($R^2$) for linear regressions of the calibration curves which were mostly > 0.99. To determine the recovery of the SE method, a mixture of standards was spiked onto the pre-cleaned blank filters which were then measured following the same procedure as for ambient samples. The recovery of the TD method was calculated based on the response signal of liquid standards and its known amount spiked onto the blank filter. Repeatability and reproducibility were also investigated for both methods. Repeatability refers to the variation in repeat measurements made under identical conditions, e.g., by the same method and the same individual while reproducibility refers to the variation in measurements made by same method but by different individuals (Bartlett and Frost, 2008). In this study, seven repeat measurements were made on n-alkanes (1000 ng $mL^{-1}$), PAHs (1000 ng $mL^{-1}$), and hopanes (1000 ng $mL^{-1}$) standards spiked on pre-baked quartz-fiber filter blank for both repeatability and reproducibility studies. The volume of these mix standards was 1 μL for the TD method and 25 μL in SE method.

For every ten samples, one replicate analysis was performed; and for every 5 samples, one backup filter was analyzed to check any potential contamination and the results show < 5% contribution. In addition, ambient samples spiked with known amounts of internal and external standards were analyzed to check the potential interference. All data reported here were corrected for the blanks.

**3 Results and discussion**

**3.1 Temperature effect on thermal desorption**

The effect of temperature in the injector port on thermal desorption is investigated by analyzing a mixture of n-alkane standards ($C_{14}$−$C_{37}$) with TD/GC-MS and compared with SE/GC-MS. The vapor pressure of these n-alkanes (molecular weight, MW, 198−520) at 275 °C ranges from 0.5 kPa to177.6 kPa, covering the vapor pressure of PAHs (0.7−97.3 kPa, MW 154−278) and hopanes (0.2−2.4 kPa, MW 370−426) investigated here. Therefore, only n-alkanes were tested to investigate the temperature effect. A range of temperature in the injector port (from 255 °C to 300 °C) were studied and the results are compared with those from the SE method (see Fig. 1). It can be seen from the figure that at lower TD temperatures (255 °C and 260 °C, see Fig. 1b and c), lighter n-alkanes have higher signal abundances than heavier ones, with the abundance of $C_{16}$ being the highest. However, the abundances of individual n-alkanes measured at these lower TD temperatures are only approximately 20% of those from the SE method, likely due to incomplete thermal desorption (Ho and Yu, 2004). With increased TD temperature (275 °C, Fig.1d), the signal abundances for all n-alkanes species increase by 2−5 times, indicating a better thermal desorption efficiency. When the TD temperature increases further to 280 °C, the signal abundances increase slightly, particularly for those lighter n-alkanes ($C_{14}$−$C_{19}$). It should be noted that the signal abundance patterns change with the TD temperature (Fig. 1b−e), with those mid-chain n-alkanes being more abundant at 275 °C and 280 °C. However, as the TD temperature further increased

to 290 °C and 300 °C (Fig. 1f and g), the signal abundances show a decreasing trend, for example, > 30% decrease in abundance for mid-chain n-alkanes at 300 °C was observed. The decrease in signal abundances at higher temperatures (e.g., 290 °C and 300 °C) is likely associated with pyrolysis. Therefore, 280 °C was selected as the optimized TD temperature to compromise for the thermal desorption efficiency and pyrolysis effect. It is noted that analytes with different molecular structure such as PAHs and hopanes may undergo different levels of pyrolysis at 280 °C when compared to n-alkanes, and may require higher/lower temperatures to get the optimized thermal desorption efficiency. However, the optimized TD temperature (i.e., 280 °C) based on the n-alkanes test was supposed to be sufficient to analyze PAHs and hopanes because similar temperature (275 °C) was used in the studies by Ho et al., (Ho and Yu, 2004; Ho et al., 2008). Also noted that even at the optimized TD temperature, the abundances of individual n-alkanes (Fig. 1e) show some differences from the SE method. In particular, the abundances of lighter n-alkanes ($C_{14}$−$C_{19}$) from TD method are about 50% lower than those from SE method, and for heavier n-alkanes ($C_{31}$−$C_{37}$) the decreases are more than 50%. However, the abundances for mid-chain n-alkanes are very similar between TD and SE methods.

### 3.2 Comparison of the analytical performance

Table 1 shows the comparison of repeatability, reproducibility, recovery, and LODs of SE/GC-MS and TD/GC-MS for n-alkanes, PAHs, and hopanes. Repeatability and reproducibility are two measures of the precision of the analytical methods. Repeatability is determined by the relative standard deviation (RSD) through measuring the identical samples (n = 7) with the same procedure while reproducibility represents the consistency between the repeated measurements by different individuals. The repeatability of SE/GC-MS is 3.5−14.5%, 4.4−10.6%, and 3.5−7.7% for n-alkanes, PAHs, and hopanes, respectively, which are similar to those obtained from TD/GC-MS (3.2–19.3%, 2.7–9.5%, and 2.1–8.4%, correspondingly). The reproducibility is also consistent for these two methods (1.2–10.9% for SE versus 1.1–12.9% for TD). In general, the precision of both SE/GC-MS and TD/GC-MS methods are satisfactory (RSD < 20%) considering the multiple sources of the errors (e.g., the error of sample pre-treatment and error from GC-MS detection).

The recoveries for the two methods were determined by analyzing backup filters spiked with known amounts of n-alkanes (1000 ng mL$^{-1}$), PAHs (1000 ng mL$^{-1}$), and hopanes (1000 ng mL$^{-1}$) standards, respectively. The recoveries of SE/GC-MS are in the range of 89.4–98.6% for n-alkanes (Table 1) which are better than those obtained from TD/GC-MS (57.2–109.8%). The good recoveries provided by SE/GC-MS indicate a good extraction efficiency and low sample matrix effect. For the TD/GC-MS method, only lighter n-alkanes ($C_{14}$−$C_{18}$) and heavier n-alkanes ($C_{32}$−$C_{37}$) show relatively lower recoveries (57.2–80.6%) (see Fig. 2), likely due to insufficient thermal desorption efficiency and/or pyrolysis effect as discussed above. The recoveries of PAHs and hopanes in TD/GC-MS method are 89.4–106.1% and 90.4–99.8%, respectively, which are as good as those obtained from SE/GC-MS method (89.7–101.5% and 90.3–97.2%, respectively).

The LODs (the concentration with a signal to noise ratio of three) of the SE method are 0.05−0.9 ng for n-alkanes, 0.3−1.1 ng for PAHs, and 0.3−1.0 ng for hopanes (Table S3). These LODs are very similar to those from the TD method, i.e., 0.09−0.8 ng for n-alkanes, 0.1−1.9 ng for PAHs, and 0.2−0.7 ng for hopanes.

### 3.3 Comparison based on ambient aerosol samples

The comparison between TD and SE methods discussed above are made based on standard compounds. When analyzing ambient samples, however, the analytical performance may be affected by e.g., matrix effect. We evaluate these potential effects by analyzing 45 ambient PM$_{2.5}$ samples collected in Beijing, Chengdu, Shanghai, and Guangzhou. Fig. 3 shows the correlation of quantitative results between SE and TD methods for n-alkanes (Fig. 3a), PAHs (Fig. 3b), and hopanes (Fig. 3c). Note that the data shown in Fig.3 are corrected for individual recoveries. The good correlations between these two methods for n-alkanes ($R^2 = 0.99$), PAHs ($R^2 = 0.98$), and hopanes ($R^2 = 0.98$) with the slopes close to unity (0.94−0.98) suggest that both TD and SE methods can be used for quantitative measurements of ambient aerosol samples.

The LODs of a method are very important for ambient measurements, particularly in regions with low concentrations or sampling with low-volume samplers. Our results show that, for example, for samples collected with a high-volume sampler at 1.1 m$^3$ min$^{-1}$ for 24 h on 8" ×10" filters, 0.53 cm$^2$ filter punch (TD method) is required to reach LODs of 0.04−0.4 ng m$^{-3}$ for n-alkanes, 0.06−0.9 ng m$^{-3}$ for PAHs, and 0.09−0.4 ng m$^{-3}$ for hopanes; and 4.3 cm$^2$ filter punch (SE method) is required to reach LODs of 1.5−27.4 ng m$^{-3}$ for n-alkanes, 9.8−33.9 ng m$^{-3}$ for PAHs, and 8.1−29.8 ng m$^{-3}$ for hopanes. In this regard, the TD method is superior to the SE method because the LODs of the TD method are about two orders of magnitude lower than the SE method and meanwhile it requires about 10 times lower sample material.

## 3.4 n-alkanes, PAHs, and hopanes in wintertime PM$_{2.5}$ in Chinese megacities

Figure 4 shows the average concentrations of individual n-alkanes species (C$_{14}$−C$_{37}$) during the winter time in Beijing, Chengdu, Shanghai, and Guangzhou. Short chain n-alkanes (C$_n$ ≤ C$_{26}$) are mainly derived from anthropogenic emissions while long chain n-alkanes (C$_n$ > C$_{26}$) are typical of the biogenic source (Xu et al., 2013).Overall, the short chain n-alkanes accounts for 72.2% (262.0 ng m$^{-3}$) of total n-alkanes in Beijing, 68.4% (246.7 ng m$^{-3}$) in Chengdu, 64.6% (111.6 ng m$^{-3}$) in Shanghai, and 55.8% (102.6 ng m$^{-3}$) in Guangzhou, indicating the major contribution of anthropogenic emissions to n-alkanes in these four cities in winter. The relatively lower contribution of short-chain n-alkanes in Guangzhou is likely due to more biogenic emissions of long-chain n-alkanes in south China than in north China during winter. The large anthropogenic contribution is supported by Cmax of n-alkane (i.e., the carbon number with maximum concentration), an indicator often used to distinguish anthropogenic from biogenic sources. N-alkanes with C$_{max}$ ≤ C$_{26}$ are mainly from anthropogenic sources while those with C$_{max}$ > C$_{26}$ are typically from biogenic sources (Xu et al., 2013). In this study, C$_{25}$ exhibits the highest concentration (66.1 ng m$^{-3}$) in Beijing while C$_{24}$ (48.0 ng m$^{-3}$) is the highest in Chengdu, C$_{22}$ (19.8 ng m$^{-3}$) in Shanghai, and C$_{26}$ (19.3 ng m$^{-3}$) in Guangzhou. We have also investigated the carbon preference index (CPI) of n-alkanes, which was calculated following the equation:

$$CPI = \frac{\sum C_{15} \text{ to } C_{37}}{\sum C_{14} \text{ to } C_{36}} \qquad (1)$$

The values of CPI ≤ 1 (or ~1) indicate that n-alkanes are from anthropogenic sources while values of CPI > 1 indicate biogenic emissions (Mancilla et al., 2016b). The CPI values of these four cities are all close to 1 (Beijing 0.9, Chengdu 0.9, Shanghai 0.8, Guangzhou 1.0), indicating that n-alkanes are mostly from anthropogenic sources (Alves et al., 2001; Mancilla et al., 2016a). It should be noted that the above discussion of anthropogenic versus biogenic sources is empirical evidence and may be subjected to relatively large uncertainties. For example, recent studies show that vehicular emissions also contain n-alkanes > C$_{26}$ (Worton et al., 2014). Moreover, the short chain n-alkanes concentrations are 1.1−2.6 times higher in Beijing than in the other 3 cities, further supporting the higher anthropogenic emissions in Beijing. This is consistent with the higher traffic fleets and larger coal usage in Beijing than in the other 3 cities studied here (Huang et al., 2014). In fact, Beijing is the only city equipped with residential heating in the four cities studied here. However, biogenic sources are also important especially in southern city Guangzhou as indicated by the relatively low short chain concentration and fraction compared to the rest three cities (Fig. 4). This is consistent with the higher vegetation coverage and temperature even during winter time (Xu et al., 2013).

The average concentrations of individual PAH species in the four cities are shown in Fig. 5. The contributions of PAH are site dependent, indicating different sources in different cities (Okuda et al., 2006; Dockery and Stone, 2007). For example, BbF is the most abundant PAH in Beijing with an average concentration of 7.3 ng m$^{-3}$ while Phe (6.9 ng m$^{-3}$) exhibits the highest concentration in Chengdu, BbF (3.5 ng m$^{-3}$) is the highest in Shanghai and BaA (3.0 ng m$^{-3}$) in Guangzhou. BbF is mainly emitted from coal combustion with high emission factor while BghiP is mainly derived from traffic emission (Katsoyiannis et al., 2011; Tobiszewski and Namieśnik, 2012). The BbF concentration (7.3 ng m$^{-3}$) in Beijing is 2−3 times as much as that in the other three cities (Chengdu 3.6 ng m$^{-3}$, Shanghai 3.5 ng m$^{-3}$, and Guangzhou 2.2 ng m$^{-3}$), supporting its large emissions

from residential coal combustion in Beijing. The BghiP concentrations Beijing and Chengdu (3–4 ng m$^{-3}$) are rather similar, indicating similar levels of traffic emission in the two cities. In addition, the BaP concentrations are similar in Beijing and Chengdu (3–4 ng m$^{-3}$) and similar in Shanghai and Guangzhou (2.0–2.5 ng m$^{-3}$). They all exceed the guideline from the World Health Organization (WHO) by a factor of 2–4, indicating potential health risk as BaP is a carcinogenic indicator.

Hopanes are pentacyclic hydrocarbons with triterpene group and can be from both coal and traffic emissions (Oros and Simoneit, 2000; Schauer et al., 2002; Zhang et al., 2008). As shown in Fig. 6, Beijing exhibits the highest concentrations of hopanes among the four cities studied. C30αβH is the most abundant hopane species in the four cities, with the highest concentration in Beijing (2.0 ng m$^{-3}$) and similar concentrations in Chengdu, Shanghai, and Guangzhou (~0.6 ng m$^{-3}$). Note that the vehicular fleets in 2014 are 5.4 million in Beijing, about 1.6−2.0 times higher than those in the other three cities (i.e., 3.4 million in Chengdu, 2.7 million in Shanghai, and 2.7 million in Guangzhou; Chinese Statistical Yearbook 2015), while the concentration of C30αβH is > 3 times higher in Beijing than in the other three cities. Such a large difference could be attributed to additional emission sources besides vehicles, that is, emissions from coal combustion for wintertime residential heating in Beijing, which is unique among the four cities we studied (Huang et al., 2014; Elser et al., 2016).

Generally, the total n-alkane concentrations rank in the order Shanghai < Guangzhou < Chengdu < Beijing, but PAHs and hopanes concentrations are in the order of Guangzhou < Shanghai < Chengdu < Beijing (Fig. 7). The average n-alkanes concentration in Beijing and Chengdu (~400 ng m$^{-3}$) are 2 times that of Guangzhou and Shanghai (~200 ng m$^{-3}$). The average concentrations of PAHs (the total of 14 PAHs) in Beijing are 46.4–47.1 ng m$^{-3}$, which are nearly two times higher than those in Guangzhou (20.8–21.3 ng m$^{-3}$), 1.7 times higher than those in Shanghai (26.2–26.7 ng m$^{-3}$), and 1.3 times higher than those in Chengdu (35.0–35.6 ng m$^{-3}$). For hopanes, the concentrations in Beijing (~6 ng m$^{-3}$) are 3 times higher than those in Chengdu, around 4 times higher than those in Shanghai and Guangzhou. As discussed above, the reasons for the higher concentrations of n-alkanes, PAHs, and hopanes observed in Beijing were attributed to emissions from traffic and/or coal combustion for wintertime heating.

## 4 Conclusion

This study evaluated and compared the analytical performances of SE/GC-MS and TD/GC-MS for the determination of n-alkanes, PAHs and hopanes by analyzing both standards and ambient PM$_{2.5}$ samples from four Chinese megacities. For the SE method, the recovery obtained is 89.4–101.5%, the limits of detection (LOD) are 0.05–1.1 ng (1.5–33.9 ng m$^{-3}$), repeatability is 3.5–14.5% and reproducibility is 1.2–10.9%, while for the TD approach, the recovery is 57.2–109.8%, the limits of detection are 0.09–1.9 ng (0.04–0.9 ng m$^{-3}$), repeatability is 2.1–19.4% and reproducibility is 1.1–12.9%. Ambient aerosol samples were simultaneously collected from Beijing, Chengdu, Shanghai, and Guangzhou during the winter of 2013 and were analyzed by the two methods. Although TD/GC-MS exhibit relatively lower recoveries for lighter n-alkanes and heavier n-alkanes, after correction the two methods show a good agreement with a high correlation efficient (R$^2$ > 0.98) and a slope close to unity. Moreover, n-alkanes, PAHs, and hopanes concentrations in wintertime are found to be much higher in Beijing than those in Chengdu, Shanghai, and Guangzhou, likely due to the vehicular emission and/or coal combustion for heating in Beijing.

*Data availability*. Raw data used in this study are archived at the Institute of Earth Environment, Chinese Academy of Sciences, and are available on request by contacting the corresponding author.

The Supplement related to this article is available online at

*Competing interests*. The authors declare that they have no conflict of interest.

Acknowledgments. This work was supported by the National Natural Science Foundation of China (NSFC) under grant No. 91644219, No. 41877408 and 4171101096), the National Key Research and Development Program of China (No. 2017YFC0212701), State Key Laboratory of Loess and Quaternary Geology, Institute of Earth Environment, CAS (SKLLQG1531) and the Cross Innovative Team fund from the State Key Laboratory of Loess and Quaternary Geology (SKLLQG) (no. SKLLQGTD1801).

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

**Table 1. Analytical performance of SE and TD/GC-MS methods for determination of n-alkanes, PAHs, and hopanes.**

| | SE method | | | | | TD method | | | | |
|---|---|---|---|---|---|---|---|---|---|---|
| | Repeatability | Reproducibility | Recovery | LOD | LOD* | Repeatability | Reproducibility | Recovery | LOD | LOD* |
| | (%) | (%) | (%) | (ng) | (ng m$^{-3}$) | (%) | (%) | (%) | (ng) | (ng m$^{-3}$) |
| n-Alkanes | 3.5–14.5 | 1.2–10.9 | 89.4–98.6 | 0.05–0.9 | 1.5–27.4 | 3.2–19.3 | 3.3–12.9 | 57.2–109.8 | 0.09–0.8 | 0.04–0.4 |
| PAHs | 4.4–10.6 | 1.6–6.5 | 89.7–101.5 | 0.3–1.1 | 9.8–33.9 | 2.7–9.5 | 4.7–8.4 | 89.4–106.1 | 0.1–1.9 | 0.06–0.9 |
| Hopanes | 3.5–7.7 | 1.2–5.8 | 90.3–97.2 | 0.3–1.0 | 8.1–29.8 | 2.1–8.4 | 1.1–4.6 | 90.4–99.8 | 0.2–0.7 | 0.09–0.4 |

*For analyzing a 0.526 cm$^2$ (TD) and 4.34 cm$^2$ (SE) punch of filters collected with high-volume samplers (sampling at 1.1 m$^3$ min$^{-1}$ or 24 h on 8" ×10" filters).

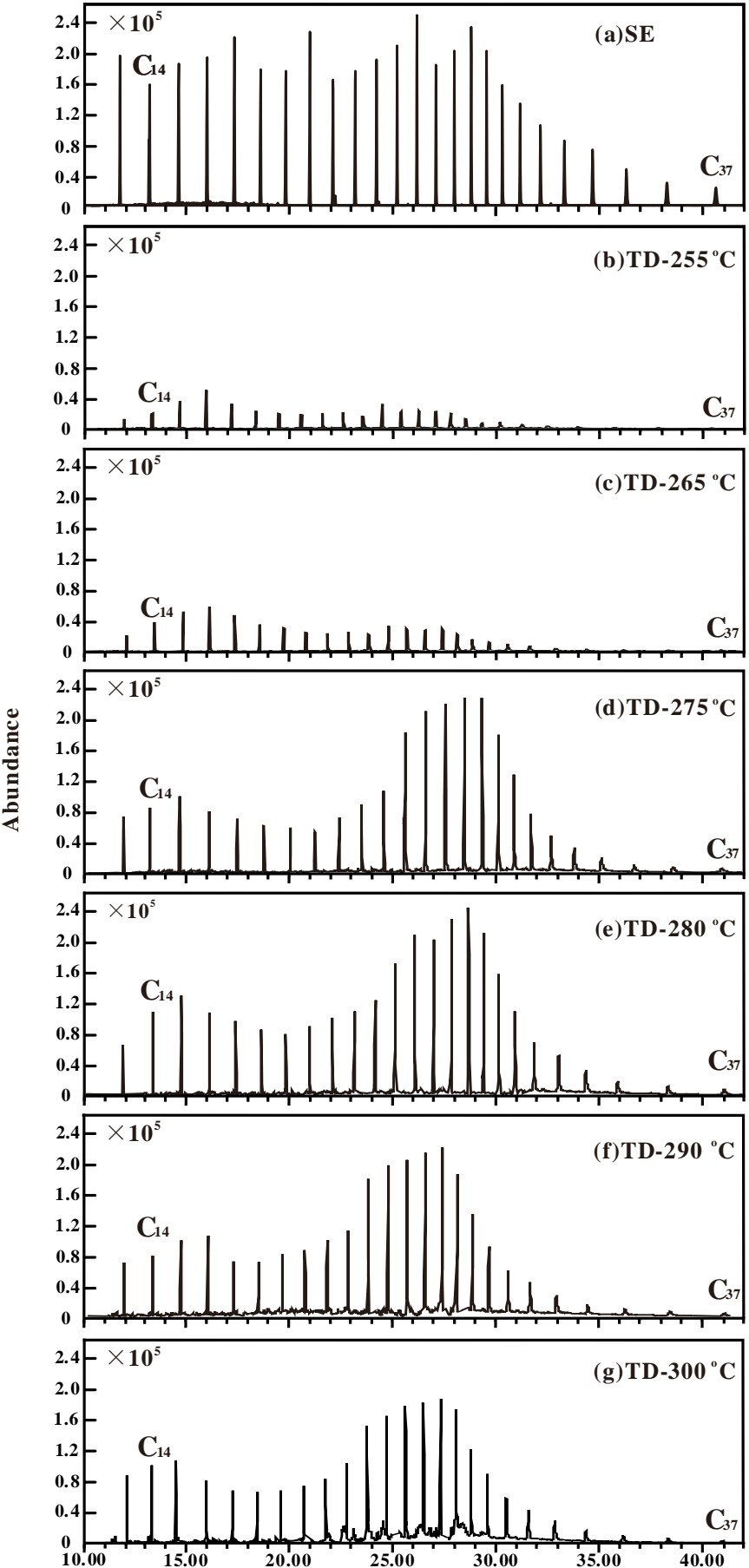

**Figure 1. Comparison of GC-MS chromatograms from solvent extraction (SE, a) and thermal desorption (TD) at different injector-port temperatures of 255 °C (b), 265 °C (c), 275 °C (d), 280 °C (e), 290 °C (f), and 300 °C (g).**

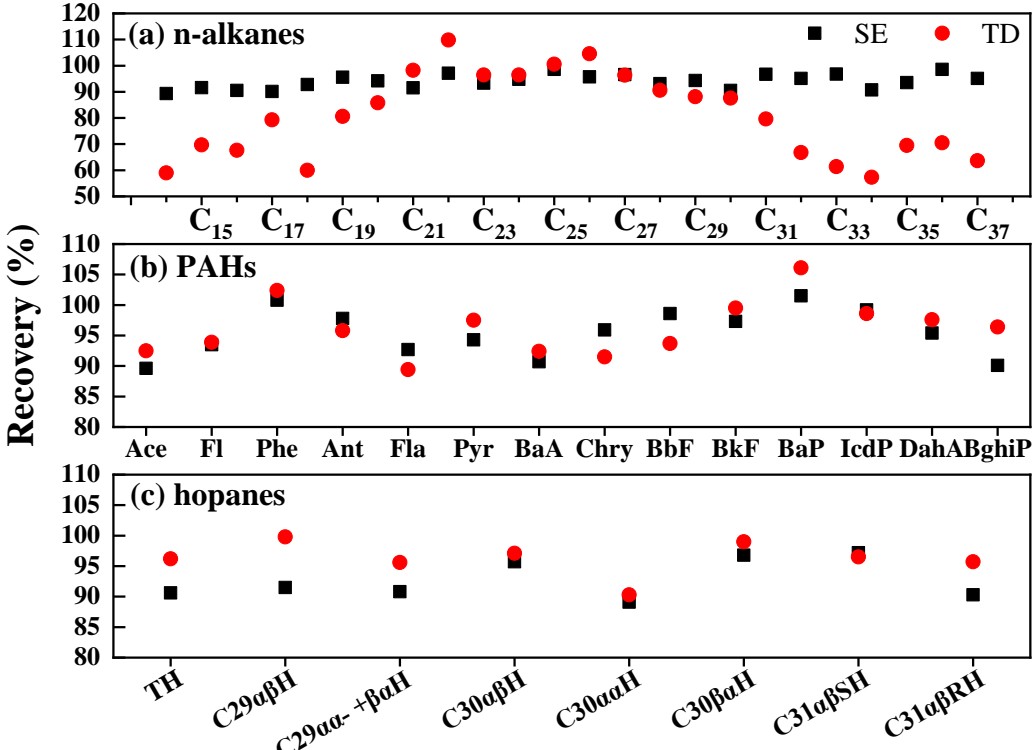

**Figure2. The recoveries of n-alkanes (a) (C_n is the straight-chain n-alkanes with carbon number from 14–37), PAHs (b) and hopanes (c) by SE and TD/GC-MS. The full names of individual abbreviations are shown in SI.**

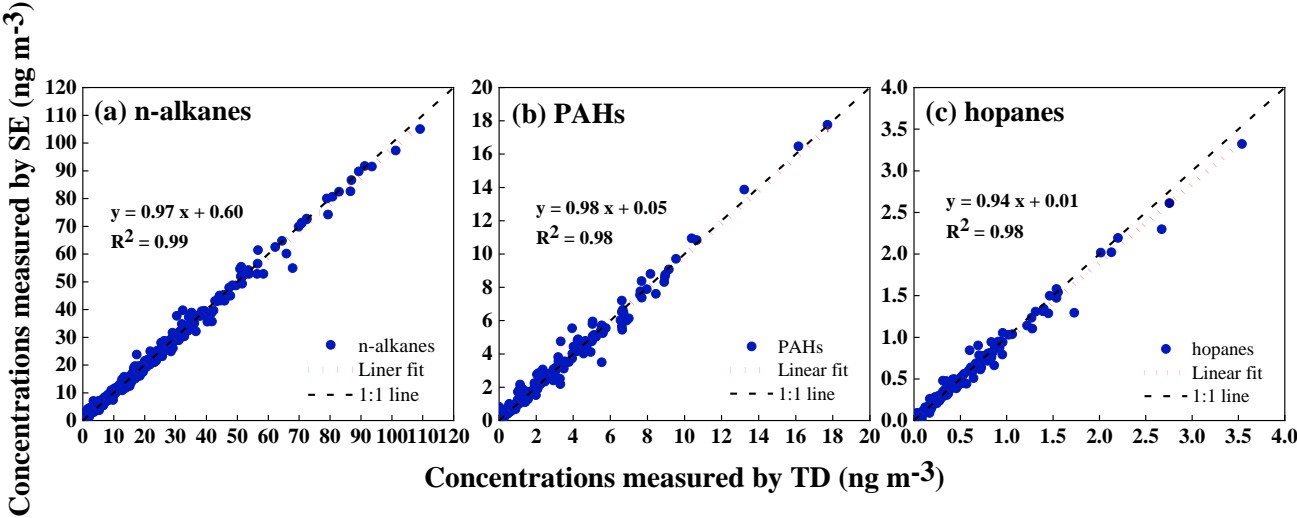

**Figure 3.** Comparisons of n-alkanes (a), PAHs (b), and hopanes (c) determined by SE/GC-MS and TD/GC-MS methods for the ambient PM$_{2.5}$ samples collected simultaneously from Beijing, Shanghai, Guangzhou, and Chengdu during winter of 2013.

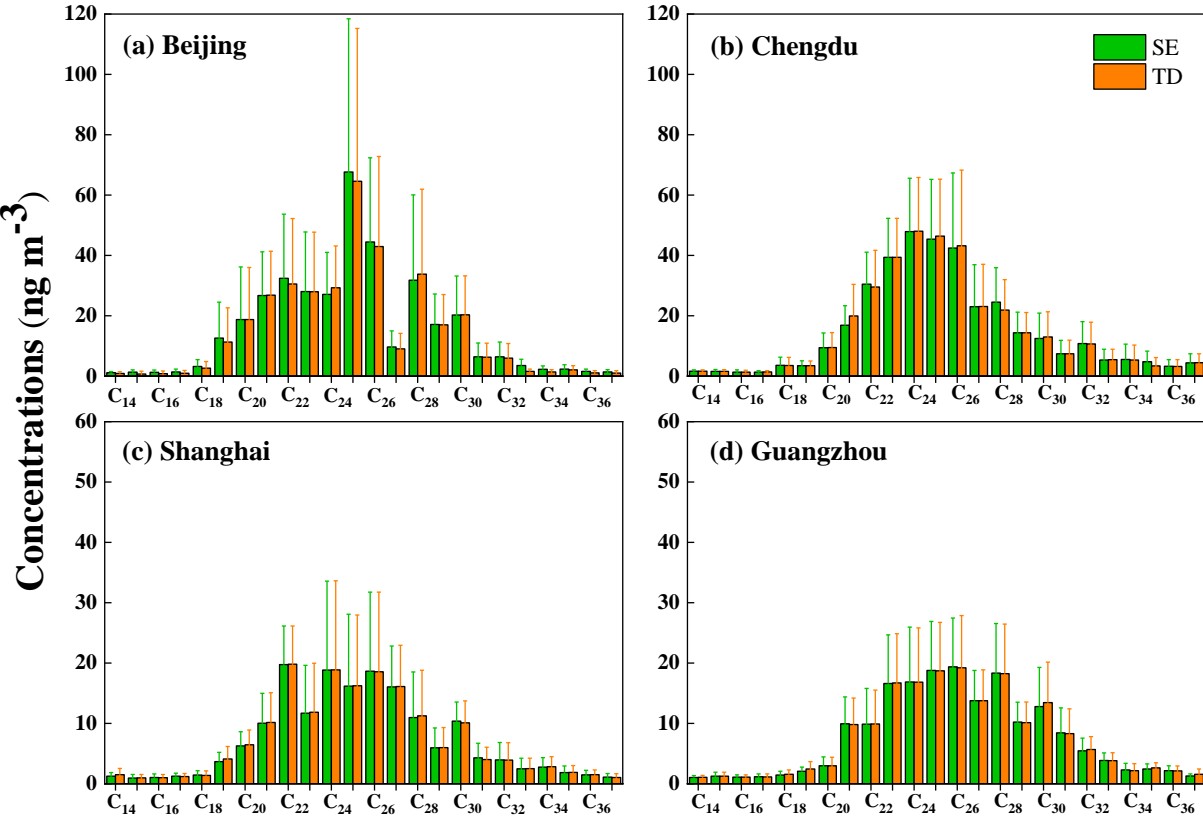

**Figure 4.** Average concentration of n-alkanes (n-C$_{14}$−n-C$_{37}$) determined by SE/GC-MS and TD/GC-MS methods during wintertime from December 2013 to January 2014 in Beijing (a), Chengdu (b), Shanghai (c), and Guangzhou (d). Error bar is one standard deviation of individual n-alkanes.

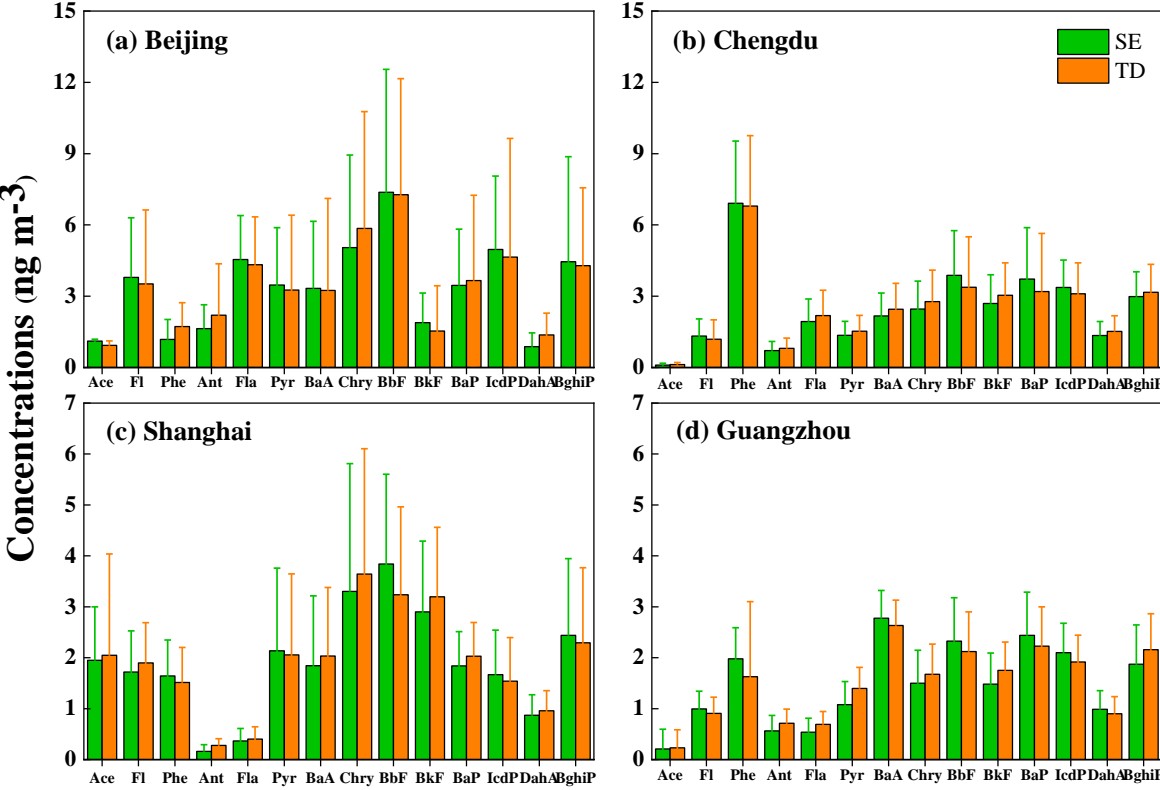

**Figure 5.** Average concentration of individual PAHs determined by SE/GC-MS and TD/GC-MS methods during wintertime from December 2013 to January 2014 in Beijing (a), Chengdu (b), Shanghai (c), and Guangzhou (d). Error bar is one standard deviation of individual PAHs.

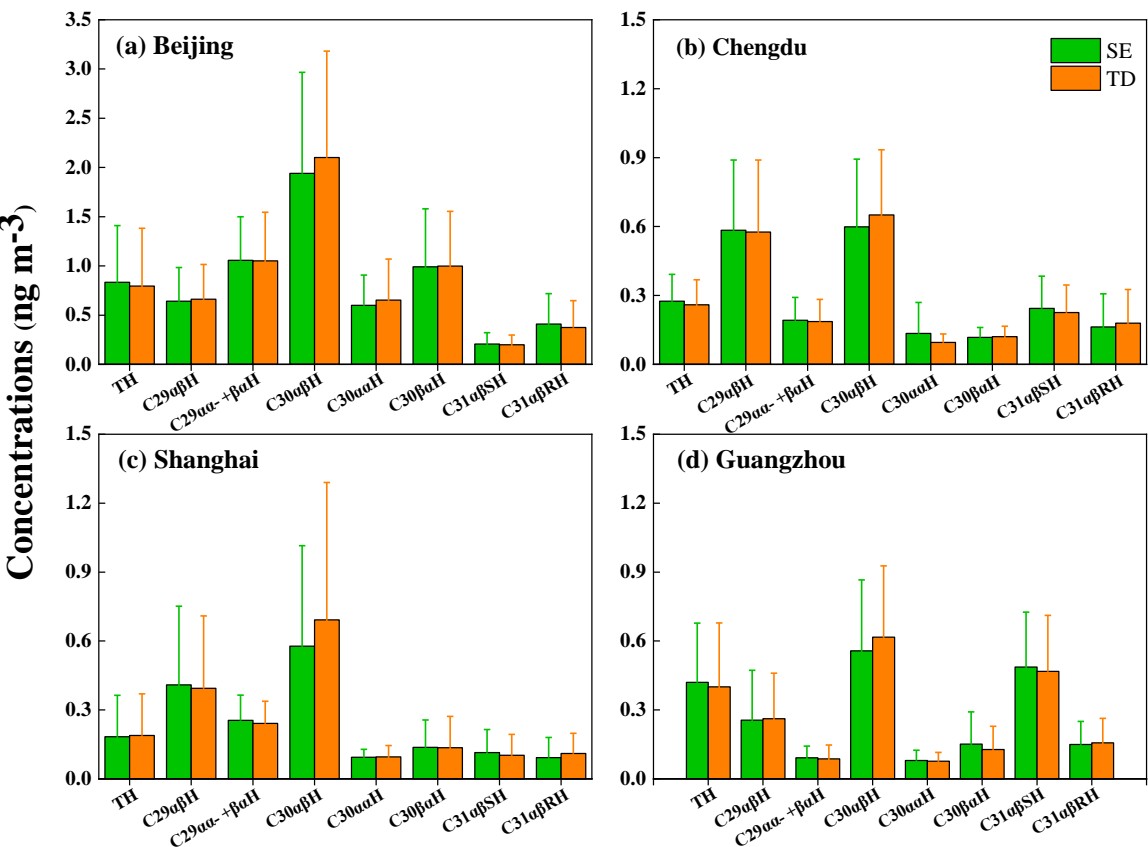

**Figure 6.** Average concentration of individual hopanes determined by SE/GC-MS and TD/GC-MS methods during wintertime from December 2013 to January 2014 in Beijing (a), Chengdu (b), Shanghai (c), and Guangzhou (d). Error bar is one standard deviation of individual hopanes.

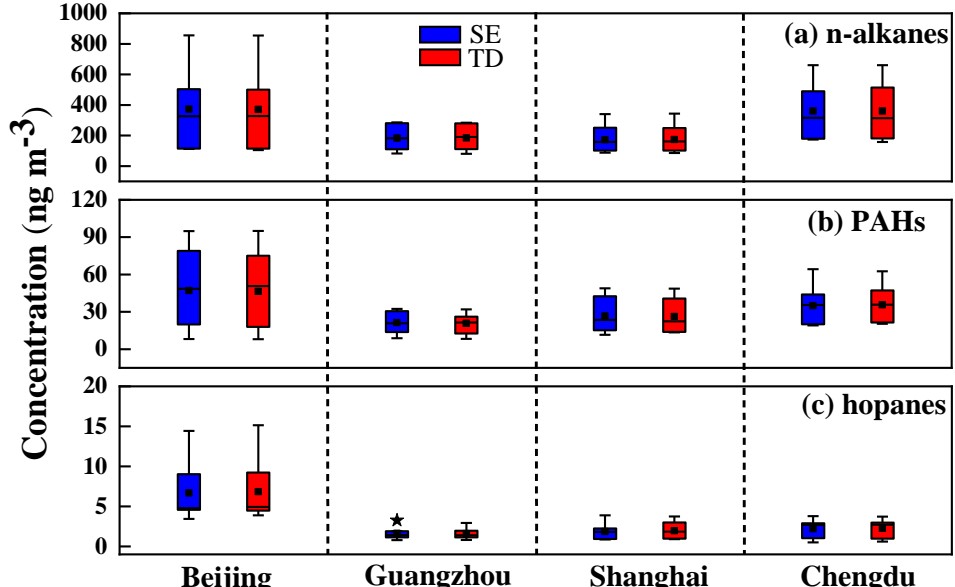

**Figure 7. Box plots of n-alkanes (a), PAHs (b), and hopanes (c) concentrations (ng m⁻³) in Beijing, Guangzhou, Shanghai, and Chengdu during winter, 2013. The boxes represent the 25th percentile (lower edge), median (solid line), mean (cube mark), and 75th percentile (higher edge). The whiskers represent the minimum and maximum. The star mark represented one outlier for the samples collected in Guangzhou.**

