# Peer review of "Determination of n-alkanes, PAHs and hopanes in atmospheric aerosol: evaluation and comparison of thermal desorption GC-MS and solvent extraction GC-MS approaches"

_Atmospheric Measurement Techniques, 2019_

## Referee Comment (RC1) · Anonymous Referee #1 · 15 Mar 2019

Line 81. The use of quartz filters is needed for easy thermal desorption analysis; however, there are many artifacts associated with these filters. The adsorption of vapors is well known, and this can affect measurements. What is the estimated contribution of this artifacts to the measurements? A set of samples with a backup filter would have provide information regarding this problem and if it contributed to differences in methods.

Line 87. There is no information regarding how the authors came up with the extraction volume, solvent mix, and sonication time. If this is a previously verified method,

there should add a reference. If is was selected after conducting preliminary extraction studies, it should be mentioned.

Line 90. There are several solvent extraction steps, and it is not clear if the suspended particles in solution were extracted with the solvent before adding new one. If this is the case considerable amounts of organics may have been removed with the particulates. This is important to consider when correcting for losses and differences in method. If particles were removed with the solvent between extraction steps, the difference between the two methods could be even greater.

Line 163. These differences indicate that the method results are dependent on the chemical properties of the sample and the n-alkanes loading, and it will need optimization for every set of samples collected in areas with different particle source profiles. Have authors test this method with samples of different nature than the ones collected in the China area? It will be important to evaluate feasibility of the analytical method for widespread use.

Line 199. The authors indicate there is a good correlation between methods once correction for the lower TD quantification. Is this realistic? A considerable analysis needs to be done to measure the efficiency of the TD method. Also, the solvent extraction method could be different, which leads to considerable number of variables that can change results. Either you always use one method, and compare samples always determined with the same issues, or evaluate differences before analysis and quantification of samples from different sources.
* * *

---

## Referee Comment (RC2) · Anonymous Referee #3 · 30 Apr 2019

The study presented in the manuscript described the comparison of two analytical method for the quantification of n-alkanes, PAHs and hopanes in atmospheric aerosol. The authors evaluated compared TD-GC/MS method with SE-GC/MS method in terms of their analytical performance and discussed their advantages/disadvantages.

The manuscript is well written but the scientific interest in comparing these two analytical methods is limited. Indeed, the study does not provide any new analytical technique and finally only the measured concentrations of particle organic compounds for samples collected in China, provide an interest to the organic aerosol community.

[Figure]

It would be necessary to improve the quality of the manuscript more discuss about the difference in aerosol sources and processes in four megacities, increase the number of sample (compare different season), interpret the data according to the meteorological parameters to the sample sites, . . .

The solvent extract method (SE) is not enough developing to make a comparaison with the thermal desoportion method. For example, other extraction method could have been tested : Soxhlet extraction or/and ASE method ; different solvent mixture (acetonitrile, . . ..), . . .

---

## Referee Comment (RC3) · Anonymous Referee #4 · 1 May 2019

Reviewer comments on amt-2019-4

The authors present a comparison between between analysis of organic particle components using solvent extraction versus thermal desorption. In general, I think it is important to do careful methodological analyses, as is the goal of this work. The overall upshot conclusion is that TD is as good a method as SE, which is a valuable conclusion. However, if the focus is on details of the comparison, there are several major shortcomings of the manuscript as it is currently presented. These are described in general comments below. If the focus is on the scientific comparison between the

cities, substantial additional conclusions and discussion would be warranted, as well as better justification for some of the current conclusions of the work (i.e. n-alkanes are not a very compelling way to distinguish biogenic an anthropogenic emissions). My overall impression of this work is that it provides good supporting information that their extraction techniques work, but falls short of providing either a comprehensive analysis of their techniques, or a compelling scientific analysis of their data. With this in mind, it is more suited for the supplementary material of a separate manuscript that does the latter.

General comments:

Gaps in methodological details: The efficiency and quality of thermal desorption is expected to be strongly dependent on the flow rates and paths of the desorbed analytes. If this is a commercially available system (e.g. Gerstel TDU, or Markes TD-100), the authors should provide that information and do not need to describe flow paths in detail. If it is custom built, some detailed description is necessary.

Was there cryogenic trapping of the desorbed analytes for focusing? Many TD systems (e.g. Gerstel and Markes) desorb at high flows to a low-volume cryo trap, backflush that trap. No mention of that approach is made here. Later data (e.g. Figure 1) indicates possible loss of higher-volatility analytes, which might be improved with a cryo trap. If one is not used in this case, I question the application of this intercomparison to many currently available TD systems.

What volume of solvent extract was actually injected? (I assume ∼1 uL)

Concerns in intercomparisons: How is "recovery" calculated? Recovery of C36 for SE is 100%, yet its peak is much lower. Given that response of an MS is non-universal and not directly related to mass from first principles, what is the SE C36 signal being compared to in order to determine it is 100%? Or is the decrease in peak height just due to peak broadening and the integrated area is the same for all alkanes? If that is the case, my suggestion for re-doing Figure 1 would make it much more clear.

[Figure]

Regarding LODs, the comparison between TD and SE is self-evident. TD analysis sends all introduced sample to the GC/MS, while SE dilutes sample to ∼1 mL, of which <1 uL (I'm assuming based on typical methods, it is never actually stated) is injected to the GC/MS. For the same amount of sample, TD should be ∼1000x more sensitive, which is what the authors find on Page 6.

Issues with scientific conclusions:

Particle emissions from vehicles also contain n-alkanes larger than C26, so it is not necessarily true those are indicative of biogenics. In fact, looking at Figure 4, all 4 cities exhibit n-alkane patterns centered around C24-C26, as would be expected for organic particles emitted from motor vehicles. Evidence for biogenic influence may be present based on odd-even ratios at Shanghai, but in general drying any anthro vs bio conclusions from the n-alkanes is questionable. Differences in n-alkane distributions may be due to differences in fuels or emissions. (Why not use pinic acid, or small acids, or other analytes, as evidence of biogenic influence?) (Worton, D. R.; Isaacman, G.; Gentner, D. R.; Dallmann, T. R.; Chan, A. W. H.; Ruehl, C.; Kirchstetter, T. W.; Wilson, K. R.; Harley, R. A.; Goldstein, A. H. Lubricating oil dominates primary organic aerosol emissions from motor vehicles. Environ. Sci. Technol. 2014, 48 (7), 3698–3706.)

A lot of discussion in the paper is of specific differences in concentrations between the cities, but there is not much context or discussion of the importance or reasons for these differences. When reasons are discussed, they are broad claims based on little data (such as the anthro v. bio discussion commented on above, or the general comments around residential fuel use in Beijing)

Technical comments:

Page 2 line 5 - Wording is a little odd. What exactly is "much less constrained"? The composition?

Page 2 lines 9-10 - The fraction of particulate matter identified depends strongly on

composition and sources. This is probably a reasonable statement in general, but it is a little narrow, non-specific, and perhaps a bit out of date. As examples of cases where this statement might not be true: - in the Amazon nearly 30% is just from 2-methyltetrols and C5-alkene triols, which are identified as specific compounds (see Hu, W. W.; Campuzano-Jost, P.; Palm, B. B.; Day, D. A.; Ortega, A. M.; Hayes, P. L.; Krechmer, J. E.; Chen, Q.; Kuwata, M.; Liu, Y. J.; et al. Characterization of a real-time tracer for isoprene epoxydiols-derived secondary organic aerosol (IEPOX-SOA) from aerosol mass spectrometer measurements. Atmos. Chem. Phys. 2015, 15 (20), 11807–11833.) - in the central valley of California most of the signal is a complex mixture of hydrocarbons that have been characterized in detail (see Chan, A. W. H.; Isaacman, G.; Wilson, K. R.; Worton, D. R.; Ruehl, C. R.; Nah, T.; Gentner, D. R.; Dallmann, T. R.; Kirchstetter, T. W.; Harley, R. A.; et al. Detailed chemical characterization of unresolved complex mixtures in atmospheric organics: Insights into emission sources, atmospheric processing, and secondary organic aerosol formation. J. Geophys. Res. Atmos. 2013, 118 (12), 6783–6796.) - In the southeastern US, ∼50% of signal can be accounted for by individual molecules that are oxidation products of individual precursors, and anoth ∼25% characterized by molecular formula. While this falls short of being "identified as specific compounds", it is much closer to complete characterization than implied by this sentence (see Zhang, H.; Yee, L. D.; Lee, B. H.; Curtis, M. P.; Worton, D. R.; Isaacman-VanWertz, G.; Offenberg, J. H.; Lewandowski, M.; Kleindienst, T. E.; Beaver, M. R.; et al. Monoterpenes are the largest source of summertime organic aerosol in the southeastern United States. Proc. Natl. Acad. Sci. 2018, 115 (9), 2038–2043.) I recommend tring to be a little more specific or add caveats to this claim (e.g. "In many environments..." or "using traditional techniques")

Page 2 lines 21-22 - many of the citations you reference above actually are comparisons between SE and TD. How does this work specifically advance the knowledge?

Page 4 line 25 - What volume of these standards?

Figure 1 - The chromatograms are interesting, but it would be much easier to intercompare if they were all on the same plots and axes. Why not overlay lines representing the integrated peak area in each case, something more like Figure 2a? Then we could intercompare and better understand differences.
* * *

---

## Referee Comment (RC4) · Anonymous Referee #2 · 10 May 2019

This manuscript presents results from an intercomparison of two analysis methods for organic aerosol, which mainly differ in the sample preparation procedure, i.e., thermal desorption (TD) and conventional solvent extraction (SE). Both methods were applied to a set of ambient aerosol samples from 4 different cities in China. The resulting chemical speciation data are interesting, but need further evaluation and discussion. As the main focus of the paper is on the comparison of the analysis methods, the conclusion of the study that the TD approach produces comparable results to the tradional SE method is a good finding, especially in light of the fact that the TD method is a

"greener" alternative. Nevertheless, various conceptual and technical issues need to be addressed by the authors prior to publication of this manuscript, as detailed below.

Specific comments:

1. Page 2, Lines 22-22: The authors state that comparison studies between TD and SE methods are still needed, but some have been reported in the literature.

2. Page 3, Lines 13-14: Did the internal standard function as both recovery and co-injection standard?

3. Page 3, Lines 14-16; Page 5, Lines 3-4: Why did the authors not use direct liquid injection for the analysis of the standards? It makes sense to have the same matrix as for actual samples analyses, but for the determination of e.g. the extraction efficiency it might have been better to directly inject the standard solutions into the GC-MS.

4. Page 3, Lines 27-28: Was such long baking time really needed for the TD tubes? And at what temperature was the glass wool baked?

5. Page 4, Lines 27-28: What type of blank was analyzed – filter, trip, field, etc. blank?

6. Page 5, Lines 3-4: Did the authors consider other factors, aside from vapor pressure, which might have influenced the temperature effect? Using n-alkanes to investigate this effect may not adequately represent the temp. effect on other compounds.

7. Page 5, Lines 15-19: The pyrolysis effect mentioned here may not apply to PAHs. In fact, such relatively low TD temperature may not be sufficient to recover higher molecular-weight PAHs. The authors may want to comment on how the temperature effect may be different for different compound classes.

8. Page 6, Lines 11-13: It would be helpful for the reader to see the individual values, which could be placed in the supplementary materials.

9. Page 7, Lines 1-6: This data interpretation is too crude! Why don't the authors report CPI values, as well as Cmax?

10. Page 7, Lines 19-20: Likewise, the conclusions from merely comparing ambient levels of these PAHs are too speculative.

11. Page 7, Lines 25-28: Why do the authors attribute these findings solely to coal combustion? How about combustion of oil? And do the authors have emission factors for coal combustion?

12. Page 8, Lines 12-13: Those high observed levels in Beijing can't be due just to coal combustion, and certainly have a significant contribution from vehicular traffic, as the city has a large vehicle fleet which is rapidly increasing.

Technical corrections:

1. Page 1, Line 32: Change "efficient" to "coefficient".

2. Page 7, Lines 12, 14: Say "PAH" rather than "PAHs".

3. Page 7, Line 32: Please, state whether these concentration numbers are for individual species.

4. Page 16, caption for Figure 2: Change "which" to "with".

---

## Author Comment (AC1) · 12 Jul 2019

We are grateful to the referees for their insightful comments. We provide below point-by-point responses to the referee's comments. We also have made most of the changes suggested by the referees in the revised manuscript.

**Anonymous Referee #1**

Line 81. The use of quartz filters is needed for easy thermal desorption analysis; however, there are many artifacts associated with these filters. The adsorption of vapors is well known, and this can affect measurements. What is the estimated contribution of this artifacts to the measurements? A set of samples with a backup filter would have provide information regarding this problem and if it contributed to differences in methods.

Response: We agree with the reviewer that adsorption of vapors might lead to sampling artifacts. We actually have investigated the potential artifacts by analyzing one backup filter for every five ambient samples and the results show < 5% contribution.

In the revised manuscript Sect. 2.4, Page 4, Lines 23-24, we have now added:

"…and for every 5 samples, one backup filter was analyzed to check any potential contamination and the results show < 5% contribution."

Line 87. There is no information regarding how the authors came up with the extraction volume, solvent mix, and sonication time. If this is a previously verified method, there should add a reference. If is was selected after conducting preliminary extraction studies, it should be mentioned.

Response: We have now added the above information and the literature in the revised manuscript. In Sect. 2.2, Page 2, Lines 39-40 to Page 3, Lines 1-2, it now reads:

"A portion (4.34 $cm^2$) of the sample or blank filter was cut with a clean stainless-steel punch, then extracted with a mixture (15 mL) of dichloromethane (DCM) (99.9%, LC grade, Mallinckrodt Laboratory Chemicals, Phillipsburg, NJ, USA) and methanol (99.9%, LC grade, Mallinckrodt Laboratory Chemicals, Phillipsburg, NJ, USA) (3:1, v/v) under ultra-sonication for 15 min and filtered through quartz wool packed in a Pasteur pipette (Fujii et al., 2016; Mohseni Bandpi et al., 2017)."

Line 90. There are several solvent extraction steps, and it is not clear if the suspended particles in solution were extracted with the solvent before adding new one. If this is the case considerable amounts of organics may have been removed with the particulates. This is important to consider when correcting for losses and differences in method. If particles were removed with the solvent between extraction steps, the difference between the two methods could be even greater.

Response: The extraction procedure used in our study has been widely used in many previous studies (e.g., Simoneit, 1999; Fujii et al., 2016; Mohseni Bandpi et al., 2017; Ahn et al., 2018). We used a mixture of dichloromethane and methanol (3:1, v/v) to cover a large range of polarities and to efficiently dissolve n-alkane, PAHs, and hopanes measured here. The mixtures of samples and solvents went through 15 min of ultrasonication in each extraction step. Therefore, even though the suspended particles (if any) were removed during the filtration between extraction steps, these targeted organics in the suspended particles should have been effectively extracted into the solvents, as confirmed by the good recoveries of the SE-method (89.4−98.6%).

In the revised manuscript in Sect. 2.2, Page 3, Lines 6-13, we have now added the following discussion:

"Note that the suspended particles (if any) were removed during the filtration between extraction steps which, however, has little influence on the extraction efficiencies of the measured organics because the extraction solvents cover a large range of polarities and can efficiently dissolve n-alkane, PAHs and hopanes measured here as demonstrated in previous studies (Fujii et al., 2016). The high extraction efficiencies are further supported by the good recoveries (89.4−98.6%) which are considered to be very satisfactory given additional sources of error (e.g., from GC-MS detection)."

Line 163. These differences indicate that the method results are dependent on the chemical properties of the sample and the n-alkanes loading, and it will need optimization for every set of samples collected in areas with different particle source profiles. Have authors test this method with samples of different nature than the ones collected in the China area? It will be important to evaluate feasibility of the analytical method for widespread use.

Response: We agree with the reviewer that the chemical properties have an influence on the TD results, as shown in Fig. 1. To evaluate the feasibility of the TD method, we therefore measured samples from 4 Chinese megacities including Beijing, Shanghai, Chengdu, and Guangzhou, representing North, East, West, and South of China where the sources of these organics are different. Moreover, we measured the samples of low-to-high mass loadings to show that the TD method can be used to measure samples of different pollution levels and different emission sources.

In the revised manuscript in Sect. 1, Page 2, Lines 27-31, we have added:

"…in ambient aerosol samples collected in four Chinese megacities including Beijing, Shanghai, Chengdu, and Guangzhou, representing North, East, West, and South of China where the sources of these organics are different. The mass loadings of PM in those four cities were also quite different. The main objective was to show that the TD/GC-MS method can be used to measure samples of different concentrations and different emission sources."

Line 199. The authors indicate there is a good correlation between methods once correction for the lower TD quantification. Is this realistic? A considerable analysis needs to be done to measure the efficiency of the TD method. Also, the solvent extraction method could be different, which leads to considerable number of variables that can change results. Either you always use one method, and compare samples always determined with the same issues, or evaluate differences before analysis and quantification of samples from different sources.

Response: For the TD method, we established the standard calibration curves for individual compounds by spiking standards of targeted compounds onto pre-cleaned filters which were then measured following the same procedure as for ambient samples. In this case, we do not correct for recoveries for TD method. For the SE method, however, because the standard calibration curves were established by direct liquid injection of the standards, we corrected for the potential loss during the sample pre-treatment.

In the revised manuscript, we have now clarified these points in Sect. 2.4, Page 4, Lines 10-12:

"Direct injection of a mixture of liquid standards with different concentrations into the SE/GC-MS system was utilized to establish a calibration curve, while a mixture of standards with different concentrations were spiked onto the pre-cleaned filters and then used to establish the calibration curves for the TD/GC-MS system."

**Anonymous Referee #2**

This manuscript presents results from an intercomparison of two analysis methods for organic aerosol, which mainly differ in the sample preparation procedure, i.e., thermal desorption (TD) and conventional solvent extraction (SE). Both methods were applied to a set of ambient aerosol samples from 4 different cities in China. The resulting chemical speciation data are interesting, but need further evaluation and discussion. As the main focus of the paper is on the comparison of the analysis methods, the conclusion of the study that the TD approach produces comparable results to the traditional SE method is a good finding, especially in light of the fact that the TD method is a "greener" alternative. Nevertheless, various conceptual and technical issues need to be addressed by the authors prior to publication of this manuscript, as detailed below.

Response: We are grateful to the referee for the positive comments on our manuscript. We have now included further evaluation and discussion on the chemical speciation and addressed the conceptual and technical issues when necessary.

Specific comments:
1. Page 2, Lines 22-22: The authors state that comparison studies between TD and SE methods are still needed, but some have been reported in the literature.

Response: We agree with the referee that a few studies have reported the comparison between TD and SE methods. For example, Ho and Yu (2004) compared these two methods for n-alkanes and PAHs with 16 ambient filter samples from Hong Kong. Ho et al. (2008) compared these two methods for alkanes, PAHs, cyclohexanes, steranes, phthalates, and hopanes with 14 ambient samples from Hong Kong and for PAHs with 19 ambient samples from Tongliang. In our study, however, we extend the comparison of these two methods to ambient samples of low-to-high concentrations from Beijing, Shanghai, Chengdu, and Guangzhou, representing North, East, West, and South of China where the sources of these organics are different. Our results show that the TD method can be used to measure samples of different pollution levels and different emission sources.

In the revised manuscript in Sect. 1, Page 2, Lines 15-20, we have now added the following discussion:

"…Yang et al., 2017). Also, Ho and Yu (2004) compared the TD and SE methods for n-alkanes and PAHs with 16 ambient filter samples from Hong Kong; and Ho et al. (2008) compared these two methods for alkanes, PAHs, cyclohexanes, steranes, phthalates, and hopanes with 14 ambient samples from Hong Kong and for PAHs with 19 ambient samples from Tongliang. However, the evaluation and comparison of TD/GC-MS with the well-established SE/GC-MS are still needed to test its capability of measuring ambient samples of low-to-high concentrations and different emission sources."

In Sect. 1, Page 2, Lines 27-31, it now reads:

"…in ambient aerosol samples collected in four Chinese megacities including Beijing, Shanghai, Chengdu, and Guangzhou, representing North, East, West, and South of China where the sources of these organics are different. The mass loadings of PM in those four cities were also quite different. The main objective was to show that the TD/GC-MS method can be used to measure samples of different concentrations and different emission sources."

2. Page 3, Lines 13-14: Did the internal standard function as both recovery and co-injection standard?

Response: Yes, these two internal standards functioned as recovery and co-injection standards.

We have clarified it in the revised manuscript in Sect. 2.2, Page 3, Lines 5-6:

"…were added as co-injection internal standards to correct for the recovery during GC-MS measurements."

3. Page 3, Lines 14-16; Page 5, Lines 3-4: Why did the authors not use direct liquid injection for the analysis of the standards? It makes sense to have the same matrix as for actual samples analyses, but for the determination of e.g. the extraction efficiency it might have been better to directly inject the standard solutions into the GC-MS.

Response: We indeed used direct liquid injections to establish the standard calibration curves for the SE method. In Sect. 2.4, Page 4, Lines 10-16, we were trying to say that we spiked a mixture of standards onto the pre-cleaned filters for determining the recovery of the SE method. For the TD method, we established the standard calibration curves for individual compounds by spiking standards of targeted compounds onto pre-cleaned filters which were then measured following the same procedure as for ambient samples.

In the revised manuscript, it now reads:

In Sect. 2.4, Page 4, Lines 10-16: "…for two methods. Direct injection of a mixture of liquid standards with different concentrations into the SE/GC-MS system was utilized to establish a calibration curve, while a mixture of standards with different concentrations were spiked onto the pre-cleaned filters and then used to establish the calibration curves for the TD/GC-MS system. Table S2 shows the slopes and the correlation coefficients ($R^2$) for linear regressions of the calibration curves which were mostly > 0.99 (Table S2). To determine the recovery of the SE method, a mixture of standards was spiked onto the pre-cleaned blank filters which were then measured following the same procedure as for ambient samples. The recovery of the TD method was calculated based on the response signal of liquid standards and its known amount spiked onto the blank filter."

In Sect. 2.3, Page 3, Lines 35-37: "…to the next analysis. The standard calibration curves were established for individual compounds by spiking standards of targeted compounds onto pre-cleaned filters which were then measured following the same procedure as for ambient samples."

4. Page 3, Lines 27-28: Was such long baking time really needed for the TD tubes? And at what temperature was the glass wool baked?

Response: Baking the TD tubes at 550 °C for at least 10 h is a standard procedure and has been used in previous studies (Ho and Yu, 2004; Ho et al., 2008; Xu et al., 2013; Wang et al., 2015). The glass wool was baked at 550 °C for at least 5 h.

We have now added the above information and the literature in the revised manuscript:

"The TD tube is a Pyrex glass tube and is cleaned following a standard procedure: it was cleaned with methanol under ultra-sonication for more than 30 min, then baked at 550 °C for at least 10 hours before use (Ho and Yu, 2004; Ho et al., 2008; Xu et al., 2013; Wang et al., 2015). A small amount of pre-baked glass wool (baked at 550 °C for at least 5 h) was used to make two plugs for holding the filter pieces in the middle of the tube."

5. Page 4, Lines 27-28: What type of blank was analyzed – filter, trip, field, etc. blank?

Response: For every 5 samples, we measured one backup filter to check any potential contamination from laboratory analysis.

We have clarified this point in our revised manuscript in Sect. 2.4, Page 4, Lines 23-24:

"…and for every 5 samples, one backup filter was analyzed to check for any potential contamination and the results show < 5% contribution."

6. Page 5, Lines 3-4: Did the authors consider other factors, aside from vapor pressure, which might have influenced the temperature effect? Using n-alkanes to investigate this effect may not adequately represent the temp. effect on other compounds.

Response: Although other factors such as the affinity of the analytes to other components in the particulate matter (sample matrix) or the filter material (filter matrix) might contribute to some extent to the retention of target analytes during thermal desorption, we believe that vapor pressure of the target analytes is still the dominating factor. As such, thermal desorption is a temperature-manifested physical process in which desorption is determined by the vapor pressure of a compound. This method has been used to determine the volatility of organics in e.g. thermodenuder and Figaero inlet of chemical ionization mass spectrometry (CIMS). Therefore, the vapor pressure is the determining factor of the temperature effect. For the selection of n-alkanes to investigate the temperature effect, the reason is that the vapor pressure range of the studied n-alkanes covers the vapor pressures of those studied PAHs and hopanes. Therefore, we believe it is appropriate by using n-alkanes to investigate the temperature effect.

7. Page 5, Lines 15-19: The pyrolysis effect mentioned here may not apply to PAHs. In fact, such relatively low TD temperature may not be sufficient to recover higher molecular-weight PAHs. The authors may want to comment on how the temperature effect may be different for different compound classes.

Response: We agree that different compound classes (e.g., PAHs and n-alkanes) may undergo different levels of pyrolysis at the same temperature as pointed out by the referee. In the revised text, we have added such comments, it now reads "It is noted that analytes with a different molecular structure such as PAHs and hopanes may undergo different levels of pyrolysis at 280 ℃ when compared to n-alkanes, and may require different temperatures to get the optimized thermal desorption efficiency. However, the optimized TD temperature (i.e., 280 ℃) based on the n-alkanes test was considered to be sufficient to analyze PAHs and hopanes and a similar temperature (275 ℃) was used in previous studies (Ho and Yu, 2004; Ho et al., 2008).

8. Page 6, Lines 11-13: It would be helpful for the reader to see the individual values, which could be placed in the supplementary materials.

Response: We agree with the reviewer and have added Table S3 in the supplementary to show the individual values.

9. Page 7, Lines 1-6: This data interpretation is too crude! Why don't the authors report CPI values, as well as Cmax?

Response: We have added CPI values together with $C_{max}$ and relevant discussion in Sect. 3.4, Page 6, Lines 13-28:

"…in Guangzhou, indicating the major contribution of anthropogenic emissions to n-alkanes in these

four cities in winter. The relatively lower contribution of short-chain n-alkanes in Guangzhou is likely due to more biogenic emissions of long-chain n-alkanes in south China than in north China during winter. The large anthropogenic contribution is supported by $C_{max}$ of n-alkane (i.e., the carbon number with maximum concentration), an indicator often used to distinguish anthropogenic from biogenic sources. N-alkanes with $C_{max} \leq C_{26}$ are mainly from anthropogenic sources while those with $C_{max} > C_{26}$ are typically from biogenic sources (Xu et al., 2013). In this study, $C_{25}$ exhibits the highest concentration (66.1 ng m$^{-3}$) in Beijing while $C_{24}$ (48.0 ng m$^{-3}$) is the highest in Chengdu, $C_{22}$ (19.8 ng m$^{-3}$) in Shanghai, and $C_{26}$ (19.3 ng m$^{-3}$) in Guangzhou. We have also investigated the carbon preference index (CPI) of n-alkanes, which was calculated following the equation:

$$CPI = \frac{\sum C_{15} \text{ to } C_{37}}{\sum C_{14} \text{ to } C_{36}} \qquad (1)$$

The values of CPI $\leq 1$ (or ~1) indicate that n-alkanes are from anthropogenic sources while values of CPI >1 indicate biogenic emissions (Mancilla et al., 2016). The CPI values of these four cities are all close to 1 (Beijing 0.9, Chengdu 0.9, Shanghai 0.8, Guangzhou 1.0), indicating that n-alkanes are mostly from anthropogenic sources (Alves et al., 2001; Mancilla et al., 2016). It should be noted that the above discussion of anthropogenic versus biogenic sources is empirical evidence and may be subjected to relatively large uncertainties. For example, recent studies show that vehicular emissions also contain n-alkanes > $C_{26}$ (Worton et al., 2014)."

10. Page 7, Lines 19-20: Likewise, the conclusions from merely comparing ambient levels of these PAHs are too speculative.

Response: In the current study, the major objective was to show that the SE/GC-MS method and the TD/GC-MS method can be used to measure samples of different concentrations and different sources. We did not provide quantitative results from in-depth source apportionment in this study to investigate on the sources of PAHs, or other species that we measured, in these different cities in China. Instead, we simply comment on, based on concentrations of PAHs measured, the potentials sources of PAHs in different cities.

11. Page 7, Lines 25-28: Why do the authors attribute these findings solely to coal combustion? How about combustion of oil? And do the authors have emission factors for coal combustion?

Response: Indeed, hopanes can be emitted from coal combustion and oil combustion. The vehicular fleets in 2014 are 5.4 million in Beijing, about 1.6−2.0 times higher than those in other three cities (i.e., 3.4 million in Chengdu, 2.7 million in Shanghai, and 2.7 million in Guangzhou). However, the concentration of the most abundant hopane species (C30αβH) is > 3 times higher in Beijing than those in the other three cities. Such a large difference could be attributed to additional emission source other than vehicles, that is, emissions from coal combustion for wintertime residential heating in Beijing, which is unique among the four cities we studied.

In the revised manuscript, we have rephrased the discussion. In Sect. 3.4, Page 7, Lines 5-13, it now reads:

"Hopanes are pentacyclic hydrocarbons with triterpene group and can be from both coal combustion and traffic emissions (Oros and Simoneit, 2000; Schauer et al., 2002; Zhang et al., 2008). As shown in Fig. 6, Beijing exhibits the highest concentrations of hopanes among the four cities studied. C30αβH is the

most abundant hopane species in the four cities, with the highest concentration in Beijing (2.0 ng m$^{-3}$) and similar concentrations in Chengdu, Shanghai, and Guangzhou (~0.6 ng m$^{-3}$). Note that the vehicular fleets in 2014 are 5.4 million in Beijing, about 1.6−2.0 times higher than those in the other three cities (i.e., 3.4 million in Chengdu, 2.7 million in Shanghai, and 2.7 million in Guangzhou; Chinese Statistical Yearbook 2015), while the concentration of C30αβH is > 3 times higher in Beijing than in the other three cities. Such a large difference could be attributed to additional emission sources besides vehicles, that is, emissions from coal combustion for wintertime residential heating in Beijing, which is unique among the four cities we studied (Huang et al., 2014; Elser et al., 2016)."

12. Page 8, Lines 12-13: Those high observed levels in Beijing can't be due just to coal combustion, and certainly have a significant contribution from vehicular traffic, as the city has a large vehicle fleet which is rapidly increasing.

Response: Agree. We meant that coal combustion could be an additional source, on top of traffic emission, of hopane in Beijing. We have rephrased this sentence. It now reads "Such a large difference could be attributed to additional emission sources besides vehicles, that is, emissions from coal combustion for wintertime residential heating in Beijing which is unique among the four cities we studied (Huang et al., 2014; Elser et al., 2016)."

Technical corrections:
1. Page 1, Line 32: Change "efficient" to "coefficient".

Response: We have changed that to "coefficient".

2. Page 7, Lines 12, 14: Say "PAH" rather than "PAHs".

Response: We have changed those to "PAH".

3. Page 7, Line 32: Please, state whether these concentration numbers are for individual species.

Response: These concentration numbers are the average concentration of total PAHs.

4. Page 16, caption for Figure 2: Change "which" to "with".

Response: We have changed that to "with".

**Anonymous Referee #3**

The study presented in the manuscript described the comparison of two analytical method for the quantification of n-alkanes, PAHs and hopanes in atmospheric aerosol. The authors evaluated compared TD-GC/MS method with SE-GC/MS method in terms of their analytical performance and discussed their advantages/disadvantages.

The manuscript is well written but the scientific interest in comparing these two analytical methods is limited. Indeed, the study does not provide any new analytical technique and finally only the measured concentrations of particle organic compounds for samples collected in China, provide an interest to the organic aerosol community.

It would be necessary to improve the quality of the manuscript more discuss about the difference in aerosol sources and processes in four megacities, increase the number of sample (compare different season), interpret the data according to the meteorological parameters to the sample sites, …

The solvent extract method (SE) is not enough developing to make a comparison with the thermal desorption method. For example, other extraction method could have been tested: Soxhlet extraction or/and ASE method; different solvent mixture (acetonitrile,….)…

Response: Thanks for your positive comment. We agree that the TD/GC-MS and SE/GC-MS methods utilized in this study are not new techniques. However, it is still of significance to evaluate/optimize the TD/GC-MS method for particulate organics and compare with the well-established SE-GC/MS method, which was the main objective of this study. Therefore, we evaluated and compared these two methods in terms of their analytical performance and discussed their advantages/disadvantages. In particular, we demonstrated that the TD/GC-MS method can be used to determine alkanes, PAHs, and hopanes in ambient aerosol samples of low-to-high concentrations and different emission sources. As we considered this manuscript is more about method comparison, we only qualitatively discuss the difference in sources of organics in these four megacities. The seasonal difference in their sources and atmospheric processes of organics in these four megacities will be discussed in a separate paper.

For the SE method, the solvent mixture (dichloromethane and methanol, 3:1, v/v) and the extraction procedure have been widely used in many previous studies (e.g., Simoneit, 1999; Fujii et al., 2016; Mohseni Bandpi et al., 2017; Ahn et al., 2018). A mixture of dichloromethane and methanol (3:1, v/v) can cover a large range of polarities and therefore effectively extract n-alkane, PAHs, and hopanes with recoveries ranging from 89.4% to 98.6%.

**Anonymous Referee #4**

The authors present a comparison between analysis of organic particle components using solvent extraction versus thermal desorption. In general, I think it is important to do careful methodological analyses, as is the goal of this work. The overall upshot conclusion is that TD is as good a method as SE, which is a valuable conclusion. However, if the focus is on details of the comparison, there are several major shortcomings of the manuscript as it is currently presented. These are described in general comments below. If the focus is on the scientific comparison between the cities, substantial additional conclusions and discussion would be warranted, as well as better justification for some of the current conclusions of the work (i.e. n-alkanes are not a very compelling way to distinguish biogenic an anthropogenic emissions). My overall impression of this work is that it provides good supporting information that their extraction techniques work, but falls short of providing either a comprehensive analysis of their techniques, or a compelling scientific analysis of their data. With this in mind, it is more suited for the supplementary material of a separate manuscript that does the latter.

Response: Thanks for pointing out the shortcomings of our work presented in this manuscript. We have improved the discussion on methodology accordingly. The main objective was to show that the TD/GC-MS method can be used to measure samples of different concentrations and different sources, which is of particularly significance for analysing organics in polluted air, for example, in China. Therefore, we analysed samples from 4 Chinese megacities including Beijing, Shanghai, Chengdu, and Guangzhou, representing North, East, West, and South of China where the sources and concentrations of these organics are different.

General comments:
Gaps in methodological details: The efficiency and quality of thermal desorption is expected to be

strongly dependent on the flow rates and paths of the desorbed analytes. If this is a commercially available system (e.g. Gerstel TDU, or Markes TD-100), the authors should provide that information and do not need to describe flow paths in detail. If it is custom built, some detailed description is necessary.

Response: For the TD unit, we replaced the commercialized GC-MS injector tube with home-made TD tube. The TD tube was a Pyrex glass tube that was home-fabricated to be 78 mm in length, 4 mm i.d., and 6 mm o.d. The length and the outside diameter were identical to those of the GC injector liner. After the loaded TD tube was placed in the injector port, the septum cap was closed and the injector-port temperature was raised from 50 °C to 280 °C for desorption (about 8−9 min). The TD system used the ultra-high purity helium (99.9999%) as the carrier gas which was held at a constant pressure of 7.7 psi and flow of 1.0 mL min$^{-1}$. The injector was set in the splitless mode for the first 2 min after the GC oven temperature program started, then switched to the split mode, and returned to the splitless mode at the end of the GC run. We have added the above information in the revised manuscript in Sect. 2.3, Page 3, Lines 27-29.

Was there cryogenic trapping of the desorbed analytes for focusing? Many TD systems (e.g. Gerstel and Markes) desorb at high flows to a low-volume cryo trap, backflush that trap. No mention of that approach is made here. Later data (e.g. Figure 1) indicates possible loss of higher-volatility analytes, which might be improved with a cryo trap. If one is not used in this case, I question the application of this intercomparison to many currently available TD systems.

Response: We did not use a cryogenic trapping unit for concentrating the analytes because the concentrations of organics analysed in this study are very high (up to 110 ng m$^{-3}$). For the TD method, the calibration curves for individual analytes were established by spiking a mixture of standards with different concentrations and internal standards onto pre-baked filters which were then analyzed directly by TD/GC-MS. Therefore, even though the loss is a bit high for some highly volatile analytes (e.g., short-chain n-alkanes), it could be still acceptable for quantitative analysis.

What volume of solvent extract was actually injected? (I assume 1 uL)

Response: Indeed, 1 μL of the extracts were injected into the GC-MS system. We have added this in the revised manuscript.

Concerns in intercomparisons: How is "recovery" calculated? Recovery of C36 for SE is 100%, yet its peak is much lower. Given that response of an MS is non-universal and not directly related to mass from first principles, what is the SE C36 signal being compared to in order to determine it is 100%? Or is the decrease in peak height just due to peak broadening and the integrated area is the same for all alkanes? If that is the case, my suggestion for re-doing Figure 1 would make it much more clear.

Response: The recoveries of individual analytes were calculated based on the response signal of a compound and its known amount spiked onto the filter. We have double-checked and found that the recovery of C$_{36}$ is 98.6%. We have re-checked the peak width for C$_{30}$ (0.126 min), C$_{31}$ (0.121 min), C$_{32}$ (0.116 min), C$_{33}$ (0.110 min), C$_{34}$ (0.116 min), C$_{35}$ (0.121 min), C$_{36}$ (0.121 min), and C$_{37}$ (0.121 min). They all show similar widths and thus the peaks were not broadened during the measurements. Actually, we found that the peak height decreased gradually from C$_{31}$ to C$_{37}$, likely due to decreasing mass spectrometric response to long-chain alkanes.

Regarding LODs, the comparison between TD and SE is self-evident. TD analysis sends all introduced

sample to the GC/MS, while SE dilutes sample to _1 mL, of which <1 uL (I'm assuming based on typical methods, it is never actually stated) is injected to the GC/MS. For the same amount of sample, TD should be _1000x more sensitive, which is what the authors find on Page 6.

Response: We agree with the reviewer that the TD method has much lower LODs because all introduced samples are measured by GC-MS. The reason we did the comparison of LODs between these two methods is to calculate the filter punch area required for the TD and SE analysis (as done in Page 6), which is useful particularly when designing sampling time/flow for samples from pristine regions, or for samples from highly polluted regions to avoid mass overloading.

Issues with scientific conclusions:
Particle emissions from vehicles also contain n-alkanes larger than C26, so it is not necessarily true those are indicative of biogenics. In fact, looking at Figure 4, all 4 cities exhibit n-alkane patterns centered around C24-C26, as would be expected for organic particles emitted from motor vehicles. Evidence for biogenic influence may be present based on odd-even ratios at Shanghai, but in general drying any anthro vs bio conclusions from the n-alkanes is questionable. Differences in n-alkane distributions may be due to differences in fuels or emissions. (Why not use pinic acid, or small acids, or other analytes, as evidence of biogenic influence?) (Worton, D. R.; Isaacman, G.; Gentner, D. R.; Dallmann, T. R.; Chan, A. W. H.; Ruehl, C.; Kirchstetter, T. W.; Wilson, K. R.; Harley, R. A.; Goldstein, A. H. Lubricating oil dominates primary organic aerosol emissions from motor vehicles. Environ. Sci. Technol. 2014, 48 (7), 3698–3706.) A lot of discussion in the paper is of specific differences in concentrations between the cities, but there is not much context or discussion of the importance or reasons for these differences. When reasons are discussed, they are broad claims based on little data (such as the anthro v. bio discussion commented on above, or the general comments around residential fuel use in Beijing)

Response: It would certainly be helpful to measure more organic markers to differentiate anthropogenic sources from biogenic sources. However, our TD system does not include an online derivatization unit and therefore is not capable of measuring polar compounds.

Following the suggestion of the reviewer, we have now added discussions about the CPIs of n-alkanes, in addition to the discussion of short-/long-chain n-alkanes and $C_{max}$. In the revised text, it now reads: "…in Guangzhou, indicating the major contribution of anthropogenic emissions to n-alkanes in these four cities in winter. The relatively lower contribution of short-chain n-alkanes in Guangzhou is likely due to more biogenic emissions of long-chain n-alkanes in south China than in north China during winter. The large anthropogenic contribution is supported by $C_{max}$ of n-alkane (i.e., the carbon number with maximum concentration), an indicator often used to distinguish anthropogenic from biogenic sources. N-alkanes with $C_{max} \leq C_{26}$ are mainly from anthropogenic sources while those with $C_{max} > C_{26}$ are typically from biogenic sources (Xu et al., 2013). In this study, $C_{25}$ exhibits the highest concentration (66.1 ng m$^{-3}$) in Beijing while $C_{24}$ (48.0 ng m$^{-3}$) is the highest in Chengdu, $C_{22}$ (19.8 ng m$^{-3}$) in Shanghai, and $C_{26}$ (19.3 ng m$^{-3}$) in Guangzhou. We have also investigated the carbon preference index (CPI) of n-alkanes, which was calculated following the equation:

$$\text{CPI} = \frac{\sum C_{15} \text{ to } C_{37}}{\sum C_{14} \text{ to } C_{36}} \qquad (1)$$

The values of CPI ≤ 1 (or ~1) indicate that n-alkanes are from anthropogenic sources while values of CPI >1 indicate biogenic emissions (Mancilla et al., 2016). The CPI values of these four cities are all close

to 1 (Beijing 0.9, Chengdu 0.9, Shanghai 0.8, Guangzhou 1.0), indicating that n-alkanes are mostly from anthropogenic sources (Alves et al., 2001; Mancilla et al., 2016). It should be noted that the above discussion of anthropogenic versus biogenic sources is empirical evidence and may be subjected to relatively large uncertainties. For example, recent studies show that vehicular emissions also contain n-alkanes > $C_{26}$ (Worton et al., 2014)."

We have added discussions about the possible reasons for the different concentrations observed in different cities. Beijing was more polluted than the other three cities mainly due to the higher vehicular fleet and the usage of coal combustion for heating while the biogenic emissions could be still importance in Guangzhou due to higher vegetation coverage and higher temperature in south China even in winter.

In Sect. 3.4, Page 7, Lines 28-34, it now reads "the short-chain n-alkanes concentrations are 1.1−2.6 times higher in Beijing than in the other 3 cities, further supporting the higher anthropogenic emissions in Beijing. This is consistent with the higher traffic fleets and larger coal usage in Beijing than in the other 3 cities studied here (Huang et al., 2014). In fact, Beijing is the only city that has centralized residential heating in the four cities studied here. In contrast, higher contribution from the biogenic source in Guangzhou is likely associated with the higher temperature and vegetation coverage in southern China, resulting in relatively low concentrations and fractions of short-chain alkanes compared to the rest three cities (Fig. 4). This is consistent with the higher vegetation coverage and temperature even during winter (Xu et al., 2013)."

We have also discussed the possible reason for the higher concentration of hopane in Beijing. Now it reads, "As shown in Fig. 6, Beijing exhibits the highest concentrations of hopanes among the four cities studied. C30αβH is the most abundant hopane species in the four cities, with the highest concentration in Beijing (2.0 ng m$^{-3}$) and similar concentrations in Chengdu, Shanghai, and Guangzhou (~0.6 ng m$^{-3}$). Note that the vehicular fleets in 2014 are 5.4 million in Beijing, about 1.6−2.0 times higher than those in the other three cities (i.e., 3.4 million in Chengdu, 2.7 million in Shanghai, and 2.7 million in Guangzhou; Chinese Statistical Yearbook 2015), while the concentration of C30αβH is >3 times higher in Beijing than in the other three cities. Such a large difference could be attributed to additional emission sources besides vehicles, that is, emissions from coal combustion for wintertime residential heating in Beijing which is unique among the four cities we studied (Huang et al., 2014; Elser et al., 2016)."

Technical comments:
Page 2 line 5 - Wording is a little odd. What exactly is "much less constrained"? The composition?

Response: The organic composition is much more complicated than the inorganic species. We have changed "much less constrained" to "much more complicated".

Page 2 lines 9-10 - The fraction of particulate matter identified depends strongly on composition and sources. This is probably a reasonable statement in general, but it is a little narrow, non specific, and perhaps a bit out of date. As examples of cases where this statement might not be true: - in the Amazon nearly 30% is just from 2- methyltetrols and C5-alkene triols, which are identified as specific compounds (see Hu, W. W.; Campuzano-Jost, P.; Palm, B. B.; Day, D. A.; Ortega, A. M.; Hayes, P. L.; Krechmer, J. E.; Chen, Q.; Kuwata, M.; Liu, Y. J.; et al. Characterization of a realtime tracer for isoprene epoxydiols-derived secondary organic aerosol (IEPOX-SOA) from aerosol mass spectrometer measurements. Atmos. Chem. Phys. 2015, 15 (20), 11807–11833.) - in the central valley of California most of the signal is a complex mixture of hydrocarbons that have been characterized in detail (see Chan, A. W. H.; Isaacman,

G.; Wilson, K. R.; Worton, D. R.; Ruehl, C. R.; Nah, T.; Gentner, D. R.; Dallmann, T. R.; Kirchstetter, T. W.; Harley, R. A.; et al. Detailed chemical characterization of unresolved complex mixtures in atmospheric organics: Insights into emission sources, atmospheric processing, and secondary organic aerosol formation. J. Geophys. Res. Atmos. 2013, 118 (12), 6783–6796.) - In the southeastern US, ~50% of signal can be accounted for by individual molecules that are oxidation products of individual precursors, and anoth ~25% characterized by molecular formula. While this falls short of being "identified as specific compounds", it is much closer to complete characterization than implied by this sentence (see Zhang, H.; Yee, L. D.; Lee, B. H.; Curtis, M. P.; Worton, D. R.; Isaacman-VanWertz, G.; Offenberg, J. H.; Lewandowski, M.; Kleindienst, T. E.; Beaver, M. R.; et al. Monoterpenes are the largest source of summertime organic aerosol in the southeastern United States. Proc. Natl. Acad. Sci. 2018, 115 (9), 2038–2043.) I recommend tring to be a little more specific or add caveats to this claim (e.g. "In many environments..." or "using traditional techniques")

Response: We agree with the reviewer that more specific statement of what is the fraction in particulate organic matter we can characterize in detail depends strongly on where we make the measurements. We have revised the sentence with respect to the studies in different environments as suggested by the referee. We have also added "in many environments" in the revised manuscript to be more specific about the claim. "Various efforts have been committed to study the composition of organic aerosol in different environments including urban areas (Chan et al., 2013) and forested areas, such as Amazon (Hu et al., 2015) and the southeastern United States (Zhang et al., 2018). However, in many environments, only 10–30% of the particulate organic matter has been identified as specific compounds despite years of effort with the most sophisticated techniques available (Hoffmann et al., 2011)."

Page 2 lines 21-22 - many of the citations you reference above actually are comparisons between SE and TD. How does this work specifically advance the knowledge?

Response: As replied to referee #2 above, we agree with the referee that a few studies have reported the comparison between TD and SE methods. For example, Ho and Yu (2004) compared these two methods for n-alkanes and PAHs with 16 ambient filter samples from Hong Kong. Ho et al. (2008) compared these two methods for alkanes, PAHs, cyclohexanes, steranes, phthalates, and hopanes with 14 ambient samples from Hong Kong and for PAHs with 19 ambient samples from Tongliang. In our study, however, we extend the comparison of these two methods to ambient samples of low-to-high concentrations from Beijing, Shanghai, Chengdu, and Guangzhou, representing North, East, West, and South of China where the sources of these organics are different. Our results show that the TD method can be used to measure samples of different pollution levels and different emission sources.

In the revised manuscript in Sect. 1, Page 2, Lines 15-20, we have now added the following discussion:

[revised manuscript text omitted]